



**Potential bioavailability of pyrogenic organic matter resembles**
**natural dissolved organic matter pools**
Emily B. Graham[+* 1,2], Hyun-Seob Song[+ 3], Samantha Grieger[1,4], Vanessa A. Garayburu-
Caruso[1,5], James C. Stegen[1], Kevin D. Bladon[6], and Allison Myers-Pigg [1,4]
[+]equal contributors
[*]Correspondence: Emily B. Graham (emily.graham@pnnl.gov)
[1] Earth and Biological Sciences Directorate, Pacific Northwest National Laboratory, Richland,
WA, USA
[2] School of Biological Sciences, Washington State University, Richland, WA USA
[3] Department of Biological Systems Engineering, Department of Food Science and Technology,
Nebraska Food for Health Center, University of Nebraska, Lincoln, NE, USA
[4] Marine and Coastal Research Laboratory, Pacific Northwest National Laboratory, Richland,
WA, USA
[5] School of the Environment, Washington State University, Richland, WA USA
[6] Department of Forest Engineering, Resources, and Management, Oregon State University,
Corvallis, OR, USA



**GRAPHICAL ABSTRACT**

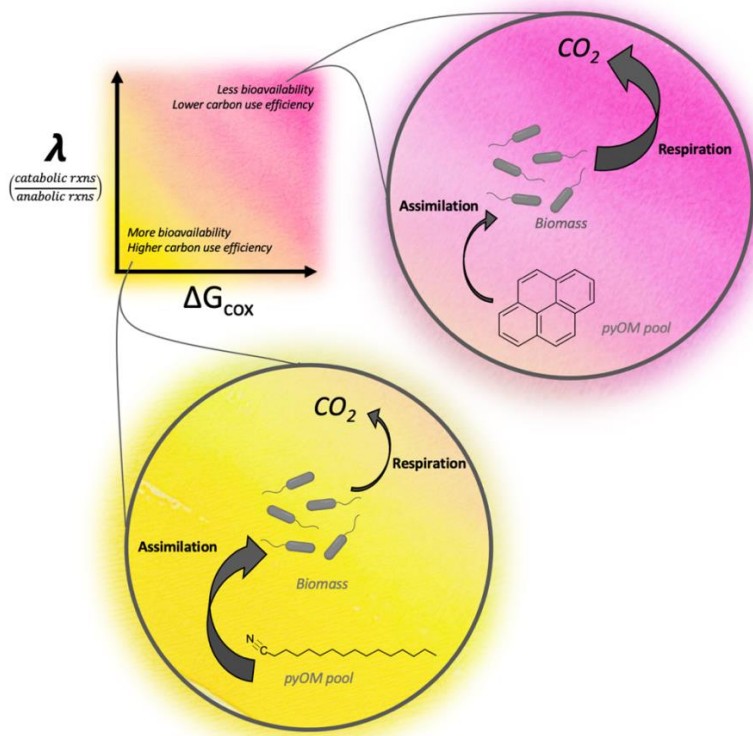



## ABSTRACT

Pyrogenic materials generated by wildfires are negatively impacting many aquatic ecosystems. At least ~10% of dissolved organic matter (DOM) pools may be comprised of pyrogenic organic matter (PyOM) that is generally considered to be more refractory than DOM from other sources. However, there has been no systematic evaluation of bioavailability across a full spectrum of PyOM chemistries. We assessed the potential bioavailability of PyOM in relation to measured and globally ubiquitous DOM compounds using a substrate-explicit model to predict the energy content, metabolic efficiency, and aerobic decomposition of representative PyOM compounds. Overall, we found similar potential bioavailability between PyOM and sediment and surface water DOM. Predicted thermodynamics and carbon use efficiencies of PyOM and DOM were statistically indistinguishable. Within PyOM, phenols and black carbon (BC, defined by Wagner et al. (2017)) had lower metabolic efficiency than other PyOM and DOM compounds, and oxygen limitation had less impact on BC metabolism than on other PyOM classes. Our work supports the recent paradigm shift where PyOM bioavailability may be more comparable to natural organic matter than previously thought, highlighting its potential role in global C emissions and providing a basis for targeted laboratory investigations into the bioavailability of various PyOM chemistries.



## 1 Introduction

Wildfires have burned an average of 1.8-million ha year$^{-1}$ in the United States alone over the past 80 years, dramatically impacting terrestrial and aquatic ecosystems (Bladon et al., 2014; Shakesby and Doerr, 2006; Randerson et al., 2006; Verma and Jayakumar, 2012). As wildfire activity continues to increase in response to climate change (Pierce et al., 2004; Bowman et al., 2020; Flannigan et al., 2009), its impact on river corridor biogeochemistry is receiving significant attention (Wagner et al., 2018; Abney et al., 2019).

Pyrogenic organic matter (PyOM) generated by wildfires, in particular, can strongly influence river corridor biogeochemistry due to the importance of organic matter as a carbon (C) and energy source in rivers. Though there is substantial uncertainty in the quantification of PyOM, estimates suggest that 116–385 Tg C is generated per year of its most common constituent—black carbon (BC: defined herein, per Wagner et al. (2017), as condensed aromatic core structures polysubstituted with O-containing functionalities). This amounts to 300 to 500 giga-metric tons of C stored in sediments, soils, and waters (Jaffé et al., 2013; Dittmar et al., 2012; Hockaday et al., 2007; Santín et al., 2016) and ~10% of dissolved organic C pools in surface waters (Jaffé et al., 2013). Given that OM drives biogeochemical cycles in most aquatic ecosystems, the loading of PyOM into river corridors has the potential to produce substantial impacts on ecosystem functions and downstream drinking water treatability (Emelko et al., 2011; Hohner et al., 2017).

Historically, PyOM has been considered refractory, passively transported and deposited throughout landscapes. While some estimates place aquatic residence times at thousands of years (Meyer and Wells, 1997; Elliott and Parker, 2001; Bigio et al., 2010; Kuzyakov et al., 2014), recent work has shown that PyOM may be more bioavailable than previously thought (Myers-





Pigg et al., 2015; Norwood et al., 2013; Zimmerman and Ouyang, 2019). This inference is also
supported by research on biochars, highlighting the diverse reactivities of combustion by-
products (Sohi et al., 2010; Mia et al., 2017). PyOM bioavailability may, therefore, play an
unrecognized role in global biogeochemical cycles and climate feedbacks. Yet, there has been no
systematic evaluation of the bioavailability of different constituents within the heterogeneous
compounds that comprise PyOM (Zimmerman and Mitra, 2017).

69          Due to the wide chemical continuum of PyOM (Wozniak et al., 2020; Masiello, 2004),

we hypothesized that a range of known PyOM compounds (i.e., from primary literature) would
show more similar potential bioavailability to a global dataset of dissolved organic matter
(DOM) pool composition than expected based on historical literature. We used a new substrate-
explicit model to assess the potential bioavailability of PyOM across its different chemical
classes and in comparison to DOM in global surface waters and sediments. The model provides a
systematic way to formulate reaction kinetics and is agnostic of many factors that have
complicated a universal understanding of OM bioavailability; including molecular structure,
chemical inhibition, mineral-associations and physical protection, terminal electron acceptors,
microbial community composition and accessibility, and abiotic reactions (reviewed in Arndt et
al. (2013)). Because it relies only on the elemental composition of individual OM molecules,
substrate-explicit modelling also enabled us to compare known PyOM compounds to detailed
characterizations of natural DOM pools that lack structural information (i.e., derived from
Fourier Transform Ion Cyclotron Resonance Mass Spectrometry, FTICR-MS). Our work
supports an emerging paradigm in wildfire science in which PyOM is relatively bioavailable and
provides a baseline for targeted laboratory experiments that examine PyOM bioavailability
across environmental contexts and compound chemistries.




## 2 Results and discussion

We used a substrate-explicit model to evaluate PyOM potential bioavailability and
compared model outputs to global DOM pool composition (Garayburu-Caruso et al., 2020a;
Song et al., 2020). In contrast to previous characterizations of PyOM bioavailability, the model-
based approach enabled us to directly compare known combustion products to thousands of
ubiquitous DOM compounds, which would have been unfeasible to directly assess in a
laboratory setting.

*2.1 Potential Bioavailability of Pyrogenic Organic Matter*

Though previous work has shown that sediment and surface water DOM is altered by
wildfires (Cawley et al., 2018; Jaffé et al., 2013; Wagner et al., 2018), our results suggest that the
chemically distinct pools of PyOM may have similar potential bioavailability to DOM. We found
that the ranges of $\Delta G_{Cox}$, $\lambda$, and CUE were similar between PyOM and DOM in sediments and
surface waters (Figure 1 and 2a). Predicted CUE of PyOM classes was also comparable to
literature values reported by others (Saifuddin et al., 2019; Domeignoz-Horta et al., 2020; Pold et
al., 2020). Similarly, $\lambda$ did not vary across groups of organic molecules (ANOVA $p = 0.09$, and
Tukey HSD $p$ (PyOM-sediment) = 0.92, $p$ (PyOM-water) = 0.40, $p$ (water-sediment) = 0.10).
While $\Delta G_{Cox}$ and CUE were significantly different when comparing all three groups (ANOVA, $p$
< 0.001), surface water and sediment DOM had greater dissimilarity in these parameters than any
comparison involving PyOM. For example, the mean difference in $\Delta G_{Cox}$ and CUE between
surface water and sediment DOM was 7.34 kJ/mol-C and 0.058. The differences between PyOM
and both surface water and sediment were less than 7.4 kJ/mol-C for $\Delta G_{Cox}$ and 0.058 for CUE.





Further, there was no evidence that CUE was different between PyOM and sediment DOM
(Tukey HSD, $p = 0.20$).

111         These results signal a strong overlap between the potential bioavailabilites of PyOM and

DOM pools; however, within PyOM compounds, there was variability in $\Delta G_{Cox}$, $\lambda$, and CUE
consistent with a heterogeneous continuum of organic matter (Figure 1 and S1). This is not
surprising, given the diversity of PyOM chemistries generated by wildfires of different burn
severities and source materials (Wagner et al., 2015; Wagner et al., 2018; Neary et al., 2005),
some of which overlap with chemical classes in unaltered DOM.

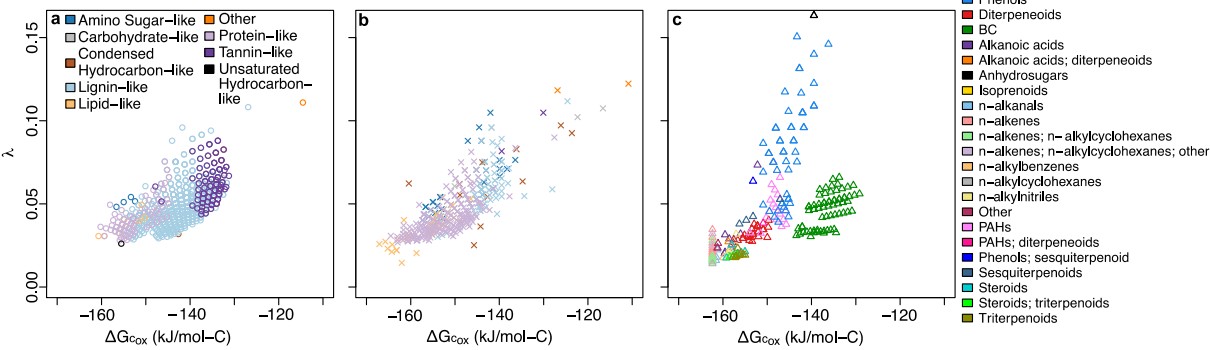



**Figure 1.** Comparison of PyOM energy content ($\Delta G_{Cox}$) and metabolic efficiency ($\lambda$) to global
DOM. Ubiquitous DOM molecules detected via FTICR-MS in global (**a**) surface water and (**b**)
sediment are colored by inferred chemical class. (**c**) Representative PyOM molecules are colored
by known chemical properties. Because PyOM molecules were from primary literature, we could
assign chemical properties at higher resolution than inferred classes from measured DOM pools.
Details on inferred chemical class assignment are provided in the Supporting Information.
Legends are inset in (**a**) for (**a**) and (**b**), and to the right of (**c**).

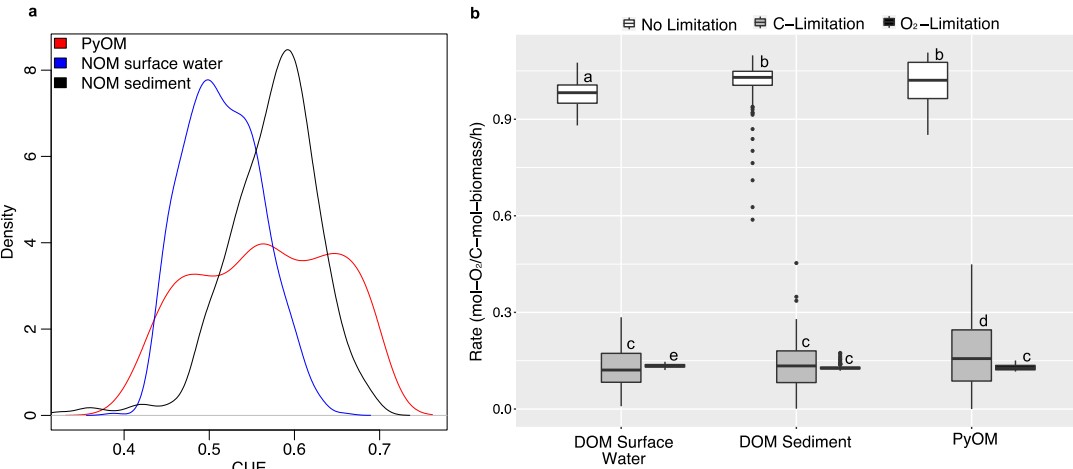

**Figure 2.** Predicted CUE and metabolism of DOM and PyOM. (**a**) shows the probability density function (PDF) of CUE in PyOM (red) and ubiquitous surface water (blue) and sediment (black). The PDF reflects the relative likelihood that value of a random sample drawn from a particular group would equal the value on the x-axis. (**b**) depicts the predicted metabolism of surface water DOM, sediment DOM, and PyOM. Letters in denote statistical groups. Median values are denoted by a bar, hinges correspond to the first and third quartiles (25th and 75th percentiles), and whiskers extend from the hinge to the largest/smallest value no further than 1.5 * IQR from the hinge (where IQR is the inter-quartile range, or distance between the first and third quartiles), and data beyond the end of the whiskers are plotted individually.

Interestingly, the relative equivalence of predicted CUE across PyOM and DOM pools suggests that PyOM decomposition in rivers could emit comparable amounts of $CO_2$ per mole of C to extant DOM pools. CUE is used in many microbially-explicit decomposition models to constrain organic matter bioavailability (reviewed in Graham and Hofmockel, 2022). Therefore,





predicted CUE offers a path for assimilating PyOM in microbially-explicit models. Such an
approach could be used to directly evaluate the impact of PyOM on global C cycles, and help
lead to better incorporation of PyOM impacts in models (Santin et al., 2020).

146         Within PyOM, two clusters of compounds were distinctly separated from the energetic

and metabolic properties of most PyOM (Figure 1c). Phenols had higher $\lambda$ than the majority of
PyOM and DOM compounds (all $\lambda > 0.039$, mean $\lambda = 0.084$), while BC molecules were less
energetically favorable than other PyOM classes (all $\Delta G_{Cox} > -143.42$, mean $\Delta G_{Cox} = -136.40$).
However, both were within the range of variability in DOM pools (Figure 1). Phenols and BC
also had among the lowest CUE (means, BC = 0.47, phenols = 0.54). Phenols are traditionally
associated with refractory organic matter, such as lignin and tannins, that exhibit long residence
times in soils (Thevenot et al., 2010), although they have also been reported to be bioavailable in
soils and waters in recent years (e.g., Thevenot et al., 2010; Ward et al., 2013). Additionally, BC
in this study is defined by inferred aromaticity (i.e., the presence of condensed aromatic
structures), which is also considered to have low reactivity (Kuzyakov et al., 2014; Wagner et al.,
2017). Although the potential bioavailability of phenols and BC is consistent with refractory
PyOM, it is within the potential bioavailability range observed in DOM, and these compounds
represent only a small portion of the PyOM continuum (Wagner et al., 2018; Masiello, 2004).
We note that the comparatively low predicted CUE of phenols and BC indicates that, if
metabolized, their decomposition could have a greater impact on river corridor $CO_2$ emissions
than other PyOM and DOM compounds. As a result, current understanding may substantially
underestimate the size, reactivity, and hydrobiogeochemical role of PyOM (Wagner et al., 2018).
*2.2 Inferred Metabolism of Pyrogenic Organic Matter.*





Predicted PyOM metabolism was also similar to DOM pools (Figure 2b), reinforcing
comparable bioavailability between the two pools. Pairwise comparison of metabolic rates
revealed no differences between PyOM and sediment DOM under oxygen limitation (Tukey
HSD, $p = 0.23$) or without C or oxygen limitations (Tukey HSD, $p = 0.34$). However, the
metabolic rates of both PyOM and sediment DOM were different than surface water DOM
(Tukey HSD, all $p < 0.001$). Aquatic sediments can reach anoxia within millimeters of the
sediment-water interface such that model predictions under oxygen limitation may translate to no
meaningful difference between PyOM and DOM in natural sediments. Under C-limitation,
PyOM had statistically elevated metabolism relative to both surface water and sediment DOM
(Tukey HSD, all $p < 0.001$). However, we noted only small differences in rate values (means,
surface water: 0.13, sediment: 0.13, PyOM: 0.17), with a similar range in sediment DOM
(0.0008–0.45) and PyOM (4.75e-08–0.45). Statistical differences were not surprising given an
extremely large sample size for DOM (sediment $n = 398$, surface water $n = 811$), and the low
effect sizes denote that overall differences in metabolism between PyOM and DOM were
minimal despite statistical separation.
When considering the impact of elemental limitations on PyOM metabolism, rate
predictions were strongly inhibited under low C and oxygen conditions. Predicted PyOM
metabolism was approximately six times lower when C or oxygen was scarce. Low
decomposition rates under C and oxygen limitation could be one reason for the observed
persistence of PyOM in depositional features that tend to be anoxic. Still, it is worthwhile to note
that metabolism of all PyOM classes under low C or oxygen was predicted to be substantially
slower than without elemental limitations, indicating PyOM compounds may both actively cycle




in well-oxygenated surface waters with fresh C inputs and persist over long periods of time in
$O_2$-limited sediments.

189         Among PyOM chemistries, BC was less negatively impacted by oxygen limitation than

any other group (Figure S2). Previous work has demonstrated that microorganisms are capable of
decomposing chemically complex organic molecules, such as long-chained and/or aromatic
hydrocarbons under low oxygen availability (Bushnell and Haas, 1941; Pozdnyakova, 2012;
Rabus et al., 2016; Coates et al., 1997). Similar microbial metabolic pathways may also be
capable of degrading BC molecules in natural settings and could be investigated with future
laboratory work. Notably, our work also supports the notion that black nitrogen could be more
bioavailable than BC. We posit this may be due, in part, to its chemical structure that includes
pyrrole-type moieties, which are relatively biodegradable (Knicker, 2010; De La Rosa and
Knicker, 2011). While we only examined one class of PyOM molecules containing nitrogen (n-
alkylnitriles), it had among the highest predicted CUE and metabolic rate.

*2.3 Correspondence to Empirical Investigations.*

202         While the substrate-explicit modelling approach used here has been validated in natural

settings and enabled comparison to environmental DOM, its underlying assumptions preclude
accounting for DOM structure and size, abiotic reactions, and chemical complexation with
minerals and particulates. Some aspects of model predictions are inconsistent with experimental
evidence, highlighting the role of laboratory studies in evaluating PyOM bioavailability. For
instance, n−alkenes and related compounds tended to have high modelled bioavailability despite
being relatively stable in the environment (including useage as paleoproxies, Wiesenberg et al.,
2004; Smittenberg et al., 2004). These compounds are characterized by carbon-carbon double



bond functional groups, which are not considered by the model and may decrease bioavailability.
However, n-alkanes generated through combustion tend to have reduced chain length in
comparison to their un-burned counterparts (Knicker et al., 2013), and thus may be relatively
bioavailable compared to un-burned n-alkanes. Additionally, we note that previous work has
shown fast degradation of combustion-derived lipids in soils (Knicker et al., 2013); as well as
high n-alkene metabolism under anaerobic conditions and in natural sediments and a range in
lipid reactivities at the sediment-water interface (Grossi et al., 2008; Wilkes et al., 2016;
Yongdong et al., 2015; Mbadinga et al., 2011; Canuel and Martens, 1996). While work on
aerobic n-alkene metabolism is limited, the comparative bioavailability of n-alkenes and known
degradation pathways suggests that sediment microbiomes may metabolize them as part of
natural biogeochemical cycles. Another notable discrepancy is the low potential bioavailability
of anhydrosugars when compared to other PyOM compounds. Experimentally, anhydrosugars
are highly bioavailable in oxic conditions, with a half-life of less than seven days (Norwood et
al., 2013). The model may therefore not adequately account for some enzyme-catalyzed
reactions such as levoglucosan kinase or levoglucosan dehydrogenase that may be common
enzymes in aquatic microorganisms (Bacik and Jarboe, 2016; Suciu et al., 2019).
Because of these nuances, the analysis presented here is best used as bounding estimates
for experimental validation and as a holistic comparison to DOM bioavailability. Still, the span
of compounds investigated here, and their comparison to DOM pools, provides a breadth of
investigation that is unfeasible without model-based approaches.

**3 Conclusions**



Our work supports the recent paradigm shift towards greater PyOM bioavailability than
previously thought and provides a foundation for targeted experiments investigating specific
components of the PyOM continuum. Globally intensifying wildfires are increasing the
production of PyOM with potential implications for source water supplies, which are critical for
domestic, industrial, agricultural, and ecological needs. Yet, many fundamental questions such as
"how much" PyOM exists in ecosystems, "how fast" it cycles, and "how old" it is remain largely
unknown (Abiven and Santin, 2019). Our work provides the first comprehensive computational
assessment of the potential bioavailabilities of various PyOM chemistries in comparison to
natural DOM pools. The comparable potential bioavailability to DOM revealed that PyOM may
be actively transformed within the river corridor and may be an increasing source of C emissions
to the atmosphere as the prevalence of wildfires increases.

**Code and Data Availability**

Code is available at: https://github.com/hyunseobsong/lambda. Data describing DOM pool
chemistry are published as a data package (Goldman et al., 2020) (available at:
doi:10.15485/1729719) and are discussed in more detail by Garayburu-Caruso et al. (2020a).

**Author Contributions**

EBG conceived of the manuscript and was responsible for writing the manuscript and generating
all figures. HSS performed all modelling. SG determined PyOM compounds for modelling based
on extensive literature review, with guidance from AMP. VGC and JCS contributed data and
insight on DOM pool chemistry. All authors contributed to revisions.




**Competing Interests**
The authors declare that they have no conflict of interest.

**Acknowledgements**
This research was supported by the U.S. Department of Energy (DOE), Office of Biological and
Environmental Research (BER), Environmental System Science (ESS) Program as part of the
River Corridor Science Focus Area (SFA) at the Pacific Northwest National Laboratory (PNNL).
PNNL is operated by Battelle Memorial Institute for the U.S. Department of Energy under
Contract No. DE-AC05-76RL01830. This study used data from the Worldwide
Hydrobiogeochemistry Observation Network for Dynamic River Systems (WHONDRS) under
the River Corridor SFA at PNNL and facilitated by the U.S. Department of Energy
Environmental Molecular Science Laboratory User Facility.

**APPENDIX. Materials and methods**
An extended version of our methods is available in the Supporting Information.
To assess the potential bioavailability of PyOM, we searched primary literature for
representative compounds of the PyOM continuum. Specifically, we targeted characteristic
organic compounds from controlled burns of various fuel types representing a range of moisture,
temperature, and oxygen conditions (Table S1). The chosen compounds focused on biomass
burning alteration products, which are often used to characterize PyOM in the environment. This
included compounds such as theoretical BC compounds, anhydrosugars, and polycyclic aromatic
hydrocarbons (PAHs). The list also included compounds created and/or transformed from
biomass burning, such as those derived from biopolymers like lignin (e.g., methoxyphenols),





waxes (e.g., n-alkenes from thermal dehydration of n-alkanols), and resins (e.g., thermally
oxidized diterpenoids) (Oros and Simoneit, 2001a, b). While we recognize that recent research
has applied new technologies to inferring PyOM compound presence in environmental samples
(e.g., FTICR-MS), there remains high uncertainty in the confidence of formula assignment and
structural information with some of these techniques. Therefore, we focused only on known,
chemically identified compounds from controlled burns to represent PyOM chemistries. The
selected set of compounds spans the chemical continuum of PyOM but was not intended to be
exhaustive. In total, our literature search for PyOM chemistries yielded 389 compounds with 207
unique chemical formulae.

287        After generating a set of representative compounds, we used a substrate-explicit

modelling framework developed by Song et al. (2020) to characterize the potential
bioavailability of each compound and predict its rate of decomposition. The model uses
molecular formulae to predict energetic content, metabolic efficiency, and rates of aerobic
metabolism, while it does not account for structural components of organic molecules (e.g.,
double bonds, folding patterns, cross-linkages). This enables flexibility in application to high-
throughput mass spectrometry techniques that yield chemical formulae but not structural
information (e.g., FTICR-MS) for comparison to environmental DOM. Despite its limitations,
the substrate-explicit model used here has proven useful in linking DOM composition to aerobic
metabolism in natural environments (Song et al., 2020; Graham et al., 2017; Garayburu-Caruso
et al., 2020b), and its structure is consistent with Harvey et al. (2016) who argued for the
importance of thermodynamic estimates of PyOM bioavailability that underlie this model. It was
chosen to allow for comparison of PyOM to the most comprehensive assessment of global
aquatic DOM pools to date (Garayburu-Caruso et al., 2020a).



Briefly, the substrate-explicit model uses the elemental stoichiometry of organic
molecules, based on molecular formulae, to predict the number of catabolic reactions that must
occur to provide the energy required for the synthesis of one mole of biomass carbon. This
quantity is described by the parameter lambda ($\lambda$) in which lower $\lambda$ values denote more efficient
energetics of catabolism in producing biomass through anabolism. The model also predicts the
Gibbs free energy of C oxidation ($\Delta G_{Cox}$), under standard conditions with a modification to pH 7
adjusted from LaRowe and Van Cappellen (2011) by Song et al. (2020), as well as C use
efficiency (CUE) as defined by Saifuddin et al. (2019). Lower $\Delta G_{Cox}$ denotes higher
thermodynamic favorability in an electron donor half reaction associated with organic matter,
and higher CUE reflects more C assimilated into biomass per unit C respired. We also predicted
the rate of aerobic metabolism (as oxygen consumed per mol-C biomass produced) under three
scenarios commonly observed in aquatic ecosystems: (a) C-limitation, (b) oxygen ($O_2$)
limitation, and (c) both C and $O_2$-limitation. For more details of the substrate-explicit modelling
approach used, please see Song et al. (2020). Each metric ($\lambda$, $\Delta G_{Cox}$, CUE, metabolic rates)
denotes a different aspect of potential bioavailability. Though the relative magnitude of the
metrics in comparison to each other will vary based on the specific stoichiometry of a molecule,
highly bioavailable compounds are indicated by low $\lambda$ and $\Delta G_{Cox}$ coinciding with high CUE and
metabolic rates.
Three sets of organic molecules were used as model inputs: measured global dissolved
(1) surface water and (2) sediment DOM pools, extracted in $H_2O$ and analyzed by FTICR-MS as
per Garayburu-Caruso et al. (2020a); and (3) literature-derived PyOM compounds as described
above. Inputs to the model from the PyOM compounds were unique molecular formulae,
grouped in subsequent analysis by their corresponding compound classes (Table S1). If one





molecular formula was represented by several PyOM compounds (e.g., $C_{10}H_{16}O_2$, which
corresponds to the sesquiterpenoid cis-Thujan-10-oic acid and 3-, 4- substituted methylcatechol
phenols), we assigned multiple compound classes to that molecular formula. Surface water and
sediment DOM pools were filtered to compounds occurring in 95% of samples to yield a dataset
of globally ubiquitous DOM. Formulae assignment and inferred chemical classes via van
Krevelen diagrams in DOM pools are described by Garayburu-Caruso et al. (2020a). We
compared modelling outputs from representative PyOM to outputs of ubiquitous DOM pools to
infer relative bioavailability using ANOVA and Tukey HSD statistical tests with R software. All
model outputs are available in Tables S2–S4.



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
