# Peer review of "compounds in comparison to natural dissolved organic matter pools"

_EGUsphere, 2022_

## Author Comment (AC2)

**Reviewer 2**

General comments

In this manuscript Graham et al. investigate the bioavailability of pyrogenic organic matter (PyOM) using a substrate-explicit model, which is then compared to that of natural dissolved organic matter and water-extracted particulate organic matter. The current understanding of the impact of PyOM in freshwaters remains mainly speculative. On this note, the manuscript addresses an important topic in riverine biogeochemistry that would be of interest to the scientific community. The manuscript is also very well-written and easy to follow. I would recommend its publication after major revisions.

Thank you for the constructive and positive comments.

Specific comments

1. Based on the compounds selected as representation of PyOM, I wonder if there is any information in the literature regarding their experimental bioavailability. The same applies to DOM and POM. The authors could expand a bit more in the introduction to further clarify the contribution of the study they are presenting.

Thank you for this comment. While empirical studies are relatively rare on PyOM bioavailability, there are several that exist (Norwood et al., 2013; Bostick et al., 2021; Chen et al., 2022). We discuss this at a high level on lines 59-68, but plan to go into more detail from these studies in the introduction.

2.      The rationale behind the experimental design is not completely clear and could not be adequate to test the proposed hypothesis. PyOM derived compounds mainly exhibit high Kow values that indicate their low solubility in water. In fact, some of the compounds included in Table S1 were determined after solvent extraction or CuO oxidation according to the references cited therein. However, these PyOM representative compounds were then compared to natural water-soluble organic matter (dissolved and particulate). Regardless, the authors report similar bioavailability parameters across phases, raising concerns about the model selection. This is because of the range of compounds with totally different chemical and physical properties that are being compared. I wonder why the list of PyOM derived compounds was not filtered to include just water-soluble compounds or the list of natural organic matter (dissolved and particulate) expanded to incorporate non-water-soluble compounds. These could represent an important overlooked fraction of natural organic matter, especially in the case of sediments. Also important is to include compounds that are detected beyond the 200-900 m/z analytical window or that escape the SPE procedure. I would recommend expanding the databases based on previously published literature and re-running the models. It would be interesting to see if similar results are obtained after expanding the composition of natural organic matter.

Per this comment and the reviewer 1 comments above, we will increase the database of PyOM molecules we are considering and will also note in the table of PyOM molecules how they were derived. If space allows, we will add in the methods section additional details on the selection choices made for the PyOM molecules – some of which currently exists in the SI – which will clarify how the approach addresses our research hypothesis.

3.      It is interesting that the authors included sediment water-extracted organic matter. This is usually not the rule in organic matter related studies in rivers, but definitely something that should be acknowledged more often.

Thank you!

4.      In the supplemental material, the authors mentioned that samples were normalised based on the concentration of dissolved organic carbon before SPE extraction. Given that the extraction efficiency of SPE cartridges is not constant, please add more information about how the organic matter extracts were normalised before FT-ICR-MS analysis or during data processing.

Thank you for this comment. We note that no new FTICR-MS data was collected for this publication. While we recognize extraction efficiencies can vary by sample, the normalization procedure used by Garayburu-Caruso et al. was intended to standardize the amount of DOM passed through each filter, which can also lead to biases. This approach has been successfully employed across a variety of published literature and is the standard operating procedure at the Environmental Molecular Sciences Laboratory. Garayburu-Caruso et al. did not report the extraction efficiencies of individual SPE cartridges.

Textor, S.R., Wickland, K.P., Podgorski, D.C., Johnston, S.E. and Spencer, R.G., 2019. Dissolved organic carbon turnover in permafrost-influenced watersheds of interior Alaska: molecular insights and the priming effect. *Frontiers in Earth Science*, *7*, p.275.

Stegen, J.C., Fansler, S.J., Tfaily, M.M., Garayburu-Caruso, V.A., Goldman, A.E., Danczak, R.E., Chu, R.K., Renteria, L., Tagestad, J. and Toyoda, J., 2022. Organic matter transformations are disconnected between surface water and the hyporheic zone. *Biogeosciences*, *19*(12), pp.3099-3110.

Danczak, R.E., Goldman, A.E., Chu, R.K., Toyoda, J.G., Garayburu-Caruso, V.A., Tolić, N., Graham, E.B., Morad, J.W., Renteria, L., Wells, J.R. and Herzog, S.P., 2021. Ecological theory applied to environmental metabolomes reveals compositional divergence despite conserved molecular properties. *Science of The Total Environment*, *788*, p.147409.

5.      Please include information regarding quality controls used during FT-ICR-MS analysis.

We note that no new FTICR-MS data was collected for this publication. We will revise the text to state that we include Suwannee River Fulvic Acid (SRFA) as a quality control check in each run in the work reported by Garayburu-Caruso et al.

6.      I would strongly suggest using the ranges proposed by Laszakovits & MacKay (2022) to assign compound classes via van Krevelen diagrams (DOI: 10.1021/jasms.1c00230). Please update.

There have been several iterations of Van Krevelen classes proposed in the literature since the original citation, and there is debate surrounding the optimal classification system. Since we did not generate any new FTICR-MS in this publication, we chose to use the classes assigned by Garayburu-Caruso et al. In response to this comment and reviewer 1, we will add Van Krevelen plots to Figure 1 and the supplemental material. One of our intents with these additions is to allow readers to examine alternate classification thresholds.

7.      Please include the F-value of the results of the statistical analysis, when appropriate, in the main body or as supplemental material.

Thank you. We will add F-values.

8.      I would recommend that the authors include a statement in the *Conclusions* addressing their previously proposed hypothesis.

Thank you. We will add a statement regarding our hypothesis in the conclusion section.

9.      The authors stated the limitations of this approach well enough (e.g., lines 226-229). This is important considering the implications and future work.

Thank you!

Technical corrections

line 53: please convert to Tg or Gg or an appropriate standard unit.

We will make this change.

line 103: please use an appropriate notation (instead of p (PyOM-sediment)).

We will make this change.

line 300: Is the dataset in Garayburu-Caruso et al. (2020a) the most comprehensive assessment of DOM in rivers to date?

We will alter the wording of this sentence to weaken this language.

line 331: please include the references for the R software as well as for each package.

We will make this change.

---

## Author Response (AR1)

Pacific Northwest
NATIONAL LABORATORY

February 5, 2023

Dear Dr. Thonicke,

Thank you for the positive comments on manuscript egusphere-2022-194, now entitled *"Potential bioavailability of representative pyrogenic organic matter compounds in comparison to natural dissolved organic matter pools"* by Emily B. Graham and colleagues.

After reviewing the comments of two anonymous reviewers, we made extensive revisions to our manuscript to address all issues raised (see point-by-point responses in the following pages). Most notably, we made the following changes:

- Added over 16,000 additional pyrogenic compounds from a variety of sources, solvents, and analytic techniques (including FTICR-MS) to generate a more comprehensive set of representative pyrogenic organic matter (PyOM)
- Revised all results, discussions, and figures accordingly including the addition of three more panels to each figure
- Revised language throughout the manuscript and in the title to provide more nuance and clarity

We attest that all authors are aware of and accept responsibility for the manuscript, and this work is original research and has not been published elsewhere. I look forward to hearing from you, and please do not hesitate to contact me with any questions.

Sincerely,

Emily B. Graham, corresponding author on behalf of co-authors

Earth and Biological Sciences Directorate, Pacific Northwest National Laboratory
School of Biological Sciences, Washington State University

**Response to Reviewer 1**

Reviewer Summary:

In this manuscript Graham et al. explore the potential of pyrogenic DOM (pyDOM) to be bio-degraded. A purely computational approach "substrate-explicit model" and previously published data were used to estimate the energy content, metabolic efficiency, and aerobic decomposition of DOM of pyrogenic and non-pyrogenic "natural" molecules. This study provides a computational explanation of why other recent studies have discovered that pyrogenic DOM can be extensively degraded (consumed) by microbes. This work contributes to the recent paradigm shift on the knowledge on pyrogenic matter's stability and reveals that pyDOM has a comparable bio-degradability (i.e., biological lability/reactivity/consumability by microbes) to natural (e.g., fluvial) DOM. This indicates that pyrogenic molecules are not as recalcitrant as previously presumed and suggesting that part of the combustion continuum is actively involved in the global biogeochemical cycles.

Reviewer Evaluation and Recommendation: This is an excellent manuscript that contributes greatly to the literature on wildfire biogeochemistry. Very well written, flowed well, smooth read, clear visuals. The bio-degradability of pyrogenic matter is currently a hot topic though there are very few studies exploring it. The present article is a great contribution to this research trajectory and is being submitted for publication (and hopefully soon published) at a great timing for the community. Indeed, as the authors mention, this and the other studies are laying the foundations of a lot of necessary future work on pyDOM microbiology/biochemistry. Computational modeling studies, especially like this one exploring bio-degradability, are generally lacking even in the "natural" DOM world, which makes this work very novel and interesting to readers from various communities.

Unfortunately, there is a major flaw with the design of the study preventing me from recommending it for publication. This flaw can heavily skew the conclusions of this work. However, this flaw can be addressed without too much trouble after which the study would be ready for publication. Thus, I recommend this paper to be published after a major revision. My other comments are minor and can be easily addressed.

*We appreciate the positivity, constructive feedback, and overall effort this reviewer put into their review. The comments were very helpful for improving the quality of the manuscript. We also appreciate the chance to address them, and we feel it resulted in a higher quality manuscript. Please see our responses to specific comments below.*

Major comment:

There are multiple concerns regarding the design of the study, in particular, the choice of natural and pyrogenic data. The conclusions of this study are heavily dependent on the comparison between these two datasets (Figure 2). Thus, complete data comparability must be ensured. Issues:

1.      Pyrogenic data (Table S1) in this study was derived from previous studies which had utilized solvent extractions. For example, a good fraction of the pyrogenic data is extracted from the studies of Oros and Simoneit who analyzed aerosol filters and extracted pyrogenic molecules using dichloromethane. Considering that in a natural environments wildfire deposits charcoal, which charcoal is then water-leached by rain, there are two immediate questions arising: 1) Is aerosol data representative of charcoal on land? 2) Is solvent-extraction comparable to water-extraction? I am unsure how comparable aerosols and charcoals are, but there is likely literature on this. However, regarding solvent versus water, a recent study showed that water and solvent extracts of charcoal are extremely different (McKenna et al., 2021). By looking at Table S1, 100% of the pyrogenic data used in this study is derived from a solvent extraction. It is highly unlikely that these solvent-extracted molecules are representative of pyrogenic molecules in aquatic environments. In order to discuss bioavailability of pyDOM in aquatic systems (the goal of this study), the pyrogenic data used here must be from water extracts. Given that both charcoals are being leached by rain, and atmospheric aerosols are deposited in waters (Wagner et al., 2021), I would say that using data of water-extracts of both charcoal and aerosols would be appropriate, but emphasis should be onto charcoal contributions as they are likely bigger.

*We recognize that listing the extraction type for our list of representative PyOM compounds may be misleading to readers and detract from our overarching goal. Our aim in generating the PyOM list was to identify molecules known to derive from pyrogenic materials, which references to are most often found in the aerosol community, although many of these molecules can also be found in natural waters and sediments (e.g. phenols, anhydrosugars, etc., Norwood et al., 2013; Suciu et al., 2019). While we included the type of extraction done and method measured, many times the solvent extractions are done primarily for preparation for the downstream analysis (e.g. GC-MS) and do not accurately reflect their water solubility (or lack thereof). While many of the references do use solvent extractions for the above reason, the molecules in question are often actually quite water soluble in natural systems (e.g. anhydrosugars, small phenols, etc; Norwood et al., 2013; Cai et al., 2020)– even molecules that are surprising based on their Kow's, due in part perhaps to their affinity and co-solubilization with bulk DOM pools (such as PAHs, Wagner et al., 2017; May et al., 1978). Therefore, while the total extractable pools in question may be quite different with solvent vs water extractions as the reviewer notes, we targeted specific molecules from a variety of pyrogenic literature sources to extract known formula that should be present beyond one analytical instruments observation window.*

*In response to this comment and similar comments below and from reviewer 2, we extensively revised our list of representative PyOM compounds. We included over 16,000 additional molecules include those derived from FTICR-MS. Our new list covers a range of solvents and source materials to present a more comprehensive evaluation. We also provide additional information in Table S1 regarding whether each compound has been identified in water soluble organic matter previously to address this comment specifically.*

*Norwood, M.J., Louchouarn, P., Kuo, L.J. and Harvey, O.R., 2013. Characterization and biodegradation of water-soluble biomarkers and organic carbon extracted from low temperature chars. Organic Geochemistry, 56, pp.111-119.*

*Suciu, L.G., Masiello, C.A. and Griffin, R.J., 2019. Anhydrosugars as tracers in the Earth system. Biogeochemistry, 146(3), pp.209-256.*

*Cai, J., Zeng, X., Zhi, G., Gligorovski, S., Sheng, G., Yu, Z., Wang, X. and Peng, P.A., 2020. Molecular composition and photochemical evolution of water-soluble organic carbon (WSOC) extracted from field biomass burning aerosols using high-resolution mass spectrometry. Atmospheric Chemistry and Physics, 20(10), pp.6115-6128.*

*Wagner, S., Ding, Y. and Jaffé, R., 2017. A new perspective on the apparent solubility of dissolved black carbon. Frontiers in Earth Science, 5, p.75.*

*May, W.E., Wasik, S.P. and Freeman, D.H., 1978. Determination of the solubility behavior of some polycyclic aromatic hydrocarbons in water. Analytical Chemistry, 50(7), pp.997-1000.*

By contrast, for "natural" DOM, authors use data fir surface waters and sediment water extracts, which data are data are appropriate for discussing aquatic biogeochemistry.

2.      Second issue identified is on the type of data used. For pyDOM, authors use known structures whereas for natural DOM authors use molecular formulas from FT-ICR-MS. The authors say that they focused their study on known pyrogenic compounds because "While we recognize that recent research has applied new technologies to inferring PyOM compound presence in environmental samples (e.g., FTICR-MS), there remains high uncertainty in the confidence of formula assignment and structural information with some of these techniques." However, the authors use FT-ICR-MS data for the natural DOM, and their own argument can be made for their natural DOM dataset. I agree that using known structures is much better, but at present we do not know most of them for pyDOM or natural DOM, so this may require the use of FT-MS (ICR or Orbitrap) molecular data for this study. Using known structures for pyDOM and using molecular formulas (from FT-ICR-MS) for natural DOM creates a major discrepancy in the comparison between pyrogenic and natural DOM. There are multiple reasons: 1) natural DOM samples were extracted with PPL (Garayburu-Caruso et al., 2020), which introduces a specific bias. 2) Only a specific fraction of molecules ionizes in negative ESI, which is the ionization used for the natural DOM dataset (Garayburu-Caruso et al., 2020). Negative ESI introduces another very specific bias onto natural DOM data. These biases are not accounted for in the structural pyDOM dataset. 3) There are entirely different data analysis routines for the natural and pyrogenic datasets (structure vs formula assignment). These routines can also possibly skew the analyzed data differently. For example, it is common to filter assigned molecular formulas following the constraints published by (Stubbins et al., 2010). If such step was done on the natural DOM data, should be also done on the pyDOM data.

*Thank you for this comment. We realize the following sentence was misleading, and we deleted it from the manuscript: "While we recognize that recent research has applied new technologies to inferring PyOM compound presence in environmental samples (e.g., FTICR-MS), there remains high uncertainty in the confidence of formula assignment and structural information with some of these techniques." As this reviewer rightly points out, this is an oversimplification of the nuances of our dataset and approach.*

*We acknowledge there are biases that result from sample preparation for FTICR-MS, as well as in the technology itself. However, it remains the gold standard for characterizing the composition of natural organic pools (Bahureksa et al. 2021). Because we are focused on inferring the bioavailability of OM, we used a reference FTICR-MS dataset of the water-soluble portion of OM which should reflect OM that is most readily available for microbial consumption (Bahureksa et al. 2021, Tfaily et al. 2017). It is well-described that compounds in water-soluble OM as detected by FTICR-MS on surface waters and sediments correspond to microbial decomposition processes (Graham et al. 2017, 2018, Garayburu-Caruso et al 2021). Therefore, we feel the NOM dataset used in this study is an adequate method to describe the composition of bioavailable OM.*

*The reviewer also correctly notes that PPL cartridges selectively remove small organic molecules and that the detectable mass range of many FTICR-MS protocols also excludes small molecules. These biases should result in a smaller range of bioavailability of OM detected by FTICR-MS than of NOM pools. Given this and the demonstrated correspondence of FTICR-MS-detected compounds with microbial processes, we feel that we can satisfactorily reach the overarching conclusion of PyOM bioavailability falling within the range of bioavailability of NOM pools.*

*We adjusted the language used in the manuscript to reflect these nuances. We also added PyOM compounds inferred from FTICR-MS to our analysis in order to provide a more comparable dataset.*

*Bahureksa, William, et al. "Soil organic matter characterization by Fourier transform ion cyclotron resonance mass spectrometry (FTICR MS): a critical review of sample preparation, analysis, and data interpretation." Environmental Science & Technology 55.14 (2021): 9637-9656.*

*Tfaily, M.M., Chu, R.K., Toyoda, J., Tolić, N., Robinson, E.W., Paša-Tolić, L. and Hess, N.J., 2017. Sequential extraction protocol for organic matter from soils and sediments using high resolution mass spectrometry. Analytica chimica acta, 972, pp.54-61.*

*Graham, E.B., Tfaily, M.M., Crump, A.R., Goldman, A.E., Bramer, L.M., Arntzen, E., Romero, E., Resch, C.T., Kennedy, D.W. and Stegen, J.C., 2017. Carbon inputs from riparian vegetation limit oxidation of physically bound organic carbon via biochemical and thermodynamic processes. Journal of Geophysical Research: Biogeosciences, 122(12), pp.3188-3205.*

*Graham, E.B., Crump, A.R., Kennedy, D.W., Arntzen, E., Fansler, S., Purvine, S.O., Nicora, C.D., Nelson, W., Tfaily, M.M. and Stegen, J.C., 2018. Multi'omics comparison reveals metabolome biochemistry, not microbiome composition or gene expression, corresponds to elevated biogeochemical function in the hyporheic zone. Science of the total environment, 642, pp.742-753.*

*Garayburu-Caruso, V.A., Stegen, J.C., Song, H.S., Renteria, L., Wells, J., Garcia, W., Resch, C.T., Goldman, A.E., Chu, R.K., Toyoda, J. and Graham, E.B., 2020. Carbon limitation leads to thermodynamic regulation of aerobic metabolism. Environmental Science & Technology Letters, 7(7), pp.517-524.*

3.      The environmental DOM dataset contains sediment water extracts. How are these relevant to aquatic microbial processes which are primarily occurring in surface waters away from sediments? I suggest removing the sediment DOM and only keeping the surficial DOM.

*While we agree that many researchers have historically focused on surface water OM, there has been increasing interest in processes that occur in subsurface hyporheic zones. These zones can contribute up to 96% of respiration in some streams (Naegeli et al. 1997). The microorganisms that catalyze OM decomposition are disproportionately bound to sediments in these zones. Therefore, we feel that sediment OM is an important consideration in the biodegradation of riverine OM. We also note that our analysis of sediment OM is separate from surficial OM and does not detract from that comparison. Based on reviewer 2's positive comments about including sediment OM, we kept this comparison in the manuscript.*

*Naegeli, Markus W., and Urs Uehlinger. "Contribution of the hyporheic zone to ecosystem metabolism in a prealpine gravel-bed-river." Journal of the North American Benthological Society 16.4 (1997): 794-804.*

4.      The environmental DOM dataset is from only one study (Garayburu-Caruso et al., 2020). The authors compare pyDOM data from multiple studies to natural DOM from one study. Is the data in Garayburu-Caruso et al. (2020) representative of multiple different aquatic environments?

*We note that the Garayburu-Caruso dataset was explicitly designed as a global survey and to capture the range of variability in global river OM. It involved a global crowdsourcing initiative that included over 97 river corridors in 8 countries within a 6-week period, from 29 July to 19 September 2019, spanning most major biomes (i.e., desert, tropical, temperate forests) and stream orders with various morphologies. All data in Garayburu-Caruso et al. were collected via standardized materials and sample procedures and analyzed by FTICR-MS at the same time, on the same instrument, using the same settings. Because of this and the challenges that this reviewer notes below in comparing FTICR-MS data from different labs with different procedures and instruments, we feel this is an excellent dataset for describing global OM pools via FTICR-MS.*

In summary, the comparison between pyrogenic and natural DOM seems like "apples to oranges" at present. All or some of the reasons listed above may skew the bioavailability of the two datasets to make them falsely appear with comparable bioavailability. This flaw needs to be addressed to be corrected in order to make the datasets comparable.

*We agree, and we acknowledge the discrepancies in comparisons. To address this, we included PyOM data derived from FTICR-MS. We note that the modelling approach we used is directly comparable across FTICR-MS and known compounds because it only considers molecular formula. Regardless of if a molecule has structural and formula information, as in the case of known molecules, structural information is not considered. This was a key factor in why this modelling approach was chosen. We also note that biases introduced by FTICR-MS should narrow the range of variability in our modelling approach, not widen it, by excluding components of NOM. The FTICR-MS procedure used by Garayburu-Caruso et al. also has been experimentally associated with biological activity. Given this, we feel that with the addition of PyOM FTICR-MS*

*data for validation, we can satisfactorily reach the overarching conclusion of PyOM bioavailability overlapping with the bioavailability of NOM pools.*

Suggestion for fixing this:

For natural DOM: add data from other studies and remove the sediment DOM. There is an overwhelming amount of FT-ICR-MS data published and provided in repositories. I am also sure that many research groups will be completely open to share data with you for this novel study. I do recommend mixing various surficial aquatic systems, primarily rivers, but also hopefully you can add lakes, wetlands, marine, etc. The choice of data will then allow you to determine if you can make claims strictly related to riverine environments or more like the global aquatic environment. Maybe you can compare pyDOM bioavailability to availability of DOM from different aquatic systems?

*We note that this is a short-format publication with a strict maximum word count of 2,500 words. This inherently limits the breadth and depth of the analysis we can perform. Because of that, we have limited this analysis to rivers only. We acknowledge that it would be very interesting to extend this analysis to other aquatic ecosystems in further work. As described above, we carefully chose our reference dataset as the Garayburu-Caruso et al. publication to have a comprehensive comparison with minimal biases as described above. For greater clarity, we provided a map of sampling locations in the SI.*

For pyDOM: I recommend using FT-ICR-MS data for ensuring comparability. Tracing truly pyrogenic molecules in natural systems is at present very challenging, so I recommend using charcoal water-extracts. There is a good number of studies that have published such data: (Chen et al., 2022; Goranov et al., 2020; Goranov et al., 2022; McKenna et al., 2021; Smith et al., 2016; Wagner et al., 2017; Ward et al., 2014; Wozniak et al., 2020; Yan et al., 2022) – just a few of the top of my citation manager.

*While we had included some suggested formulae in the previous manuscript extracted from tables in published works (n = 67 from Hockaday et al., 2006; Wagner et al., 2017), we agree that this paper would benefit from additional FTICR-MS datasets for the PyOM comparison. Thank you for the useful citations. We pulled vegetation char-derived spectra from these papers as well as several others. This resulted in over 16,000 total PyOM molecules that we analyzed for the revision. Sources for each molecule are listed in Table S1.*

I think by doing this you will achieve complete comparability (ensuring data were from PPL extracts and -ESI). I can foresee one issue – if obtaining data from multiple groups, you might get molecular formulas which could be biased by the different software that groups use (ICBM, pyKrev, Formularity, etc.). What I suggest is inquiring for peak lists (m/z and intensity data) and you work up the data yourselves to avoid comparability issues from different processing routines.

*Thank you for this suggestion! While we were unable to obtain peak lists, and as this reviewer is aware, combining peaks from different FTICR-MS instruments with different calibrations and ion*

*accumulation times is problematic even if consistent computational approaches are applied. To address this comment, we greatly expanded the range of PyOM compounds we investigated to over 16,000 molecules. We feel this expansion is satisfactorily describes the range of compounds contained in PyOM, and we do not attempt to make comparisons between individual molecules in our datasets.*

Detailed Review and Specific Comments:

Abstract: Excellent. Gives a comprehensive overview of the study, information is succinctly presented. I only recommend adding one sentence somewhere around lines 29-31 that explains that this model is a computational approach using molecular formula data from mass spectrometry or molecular formulas of known DOM structures. Readers who are not familiar with this "substrate-explicit model" will likely be confused, so enhancement in clarity is needed.

*We have substantially revised the abstract to reflect changes to the manuscript.*

Intro: Very good. Establishes the importance of wildfires and pyrogenic DOM, provides background on the modeling approach and authors identify the gaps in our knowledge of pyrogenic DOM. Authors also establish a clear objective for the study. Some minor revisions are needed:

Terminology throughout the manuscript:
- Line 45: River corridor biogeochemistry. From the abstract and title of the paper it seems that the study is going to make claims regarding broad DOM across various aquatic systems, not just fluvial ones. I suggest replacing with "aquatic biogeochemistry" throughout the manuscript.

*Please see our response above. We are strictly limited to 2,500 words for this article type. While expanding this analysis to all aquatic systems would be very interesting, it is beyond the scope we can address in this article.*

- pyOM is commonly used for particulate OM whereas pyDOM is used for dissolved OM. I advise using pyDOM as using pyOM for an aquatic study is confusing.

*We recognize that the nomenclature surrounding pyrogenic organic matter is quite varied and often misleading (Zimmerman and Mitra, 2017). Because our PyOM reference set includes a range of source materials, we have chosen to retain the broader 'PyOM' term vs. 'PyDOM'.*

*Zimmerman, A.R. and Mitra, S., 2017. Trial by fire: on the terminology and methods used in pyrogenic organic carbon research. Frontiers in Earth Science, 5, p.95.*

- This term needs to be more clearly defined. Authors should also consider using "bio-lability" as one may argue that all molecules surrounding microbes are available to the microbes, but only some molecules can be uptaken and consumed (i.e., labile to bio-degradation).

*Thank you for this comment. The term 'bioavailability' is used throughout biogeochemical literature to include both microbial access to OM and the ability for it to be consumed. We recognize biolability is preferred by some fields of research, so we included both terms at first*

*mention along with a clear definition of 'bioavailability'. Please see lines 60-61: "bioavailable (i.e., biolabile or able to be accessed and degraded by microorganisms)".*

- BC – having BC and pyOM in the same manuscript is confusing as many readers view them as synonymous. I recommend removing BC entirely. In studies employing molecular data (from FT-ICR-MS, etc.) using a carbon term is also not appropriate as the data is reported for the whole molecules (i.e., on matter-basis) and not just for the carbon backbone (i.e., not on a carbon-basis). In simple terms, FT-ICR-MS measures DOM, not DOC. I recommend using terms that directly correspond to the structure (i.e., condensed aromatic compounds, ConAC, polycyclic aromatics, PCA, oxygenated PAHs, OPAH, or others that are used among different research groups).

*Thank you for this comment. As outlined above, we recognize the considerable variability in terms used for pyrogenic derived organic matter and the need for more standardized vocabulary across studies. We agree with the reviewer that BC is not synonymous with PyOM, yet many readers may assume that it is. Therefore, we have removed the terms 'BC' and 'black carbon' from the manuscript.*

*By adding in FTICR-MS datasets, we do not feel that we can assign structural information to pyrogenic assignments from this technique as the reviewer is suggesting, but we do agree we can be more specific with the continuum of the types of molecules this diverse pool contains. We modified language throughout the manuscript with these comments in mind.*

- "Natural" DOM. I recognize the necessity to use this term, the problem with it is that some pyrogenic molecules exist in detectable quantities in natural DOM (Goranov et al., 2022). Or they share the same molecular formulas but are different isomers. Maybe authors should acknowledge this and clearly state in the intro that natural DOM corresponds to environmentally ubiquitous molecules obtained from fieldwork whereas the pyDOM dataset is from laboratory experiments and extractions to ensure these molecules are truly pyrogenic. One side question – is there any overlap (i.e., common formulas) among the natural and pyrogenic DOM datasets that the authors used? Are those common formulas removed or they were kept in? Please clarify this in the methods section.

*We agree that 'natural' DOM sources may include pyrogenic materials. For our 'natural' DOM datasets, we used compounds present in at least 95% of surface waters or sediments to minimize variation caused by molecules found in very specific environments (e.g., post-wildfire). In the revision of our introduction, we now provide details on the natural DOM data in the introduction (lines 70-72): "Natural DOM pool composition was derived from a recent high-resolution survey of river corridor sediments and surface waters (Garayburu-Caruso et al., 2020a, Figure S1)."*

*Most of our datasets were not redundant. We now report overlaps on lines 334-336: "As a whole, 16,332 compounds were found only in PyOM, 197 were found only in surface water DOM, and 167 were found only in sediment DOM." We also provide Figure S2 to show van Krevelen diagrams of unique molecules in each subset of data.*

- Lines 66-68. Authors make the claim that "Yet, there has been no systematic evaluation of the bioavailability of different constituents within the heterogeneous compounds that comprise PyOM". This was actually recently done by two separate research groups (Bostick et al., 2021; Chen et al., 2022; Goranov et al., 2022) who looked at various constituents of pyrogenic DOM using various analytical approaches. Those studies also use "true" pyDOM from biochar extracts similar to Norwood et al. 2013, but of more wildfire-representative temperatures.

*We agree. We deleted this sentence and replaced with the following: "Yet, we are just beginning to understand its potential bioavailability (Zimmerman and Mitra, 2017; Wozniak et al., 2020; Masiello, 2004)."*

- Lines 69-72. I think it would be good to explain the reason for this discrepancy – most research to date assumed that pyOM/pyDOM are only comprised of condensed structures, and the latter are indeed bio-refractory (Bostick et al., 2021), but the rest of pyOM/pyDOM is likely not.

*Thank you for this suggestion. We significantly revised this entire paragraph to include more information from the methodology. Please see lines 67-82.*

- Please expand on the substrate-explicit model text after line 82. This is the first time to my knowledge that this modeling approach is used in the wildfire biogeochemistry literature – most readers will have no background in it. Tell us about the model output: on line 99 you mention $\Delta$GCox, $\lambda$, and CUE for the first time without providing any info on what they are. There is a lot of great text in the materials and methods that does not read like M&M text – move some of it to the intro to provide a foundation on what these parameters are.

*We reorganized the manuscript to move some of the information from the M&M about modelling into the introduction and the beginning of the results and discussion. Please see lines 67-97.*

Results and Discussion: Excellent. Results are properly discussed in the context of previously existing literature. Several comments:

- We know that behind a m/z value (or a molecular formula) there could be multiple isomers with the same elemental make-up (Leyva et al., 2019). Different isomers will have different thermodynamic properties. Do the mathematics behind the model consider this? This needs to be discussed in the last section where authors discuss the limitations.

*We agree that there are often multiple isomers with the same formula. The modelling approach we used was designed specifically with this limitation of FTICR-MS data in mind. It only considers molecular formula and does not incorporate structural information. To our knowledge, the is the only extant modelling approach to predict aerobic metabolism rates from FTICR-MS data. It is now incorporated into the US Department of Energy's KBase modelling ecosystem and os seeing increasing usage for providing meaningful predictions from FTICR-MS data in other contexts.*

*Due to the short format of the article, we are unable to go into extensive details on the interworkings of the modelling approach in the main text of the manuscript. We revised the manuscript to concisely state its strengths and weaknesses in the introduction on lines 72-80: "The substrate-explicit model used molecular formulae to predict energetic content, metabolic efficiency, and rates of aerobic metabolism for organic substrates. It has proven useful in linking*

*DOM composition to aerobic metabolism in natural environments (Song et al., 2020; Graham et al., 2017; Garayburu-Caruso et al., 2020b). It provides a systematic way to formulate reaction kinetics and is agnostic of many factors that have complicated a universal understanding of OM bioavailability; including molecular structure, chemical inhibition, mineral-associations and physical protection, terminal electron acceptors, microbial community composition and accessibility, and abiotic reactions (reviewed in Arndt et al. (2013))."*

*We also dedicate an extensive portion of the methods and supplemental material to provide more details on the modelling methodology.*

Does the model account for potential toxicity of molecules? Some pyrogenic molecules are toxic (Smith et al., 2016), but not sure if a mathematical proxy for toxicity can be extrapolated from previous studies and incorporated into your computation approach. This is probably something to consider and discuss in your last section too.

*Toxicity is not considered in the model. It is an interesting topic but is beyond the scope of this paper. Please refer to the supplemental material for a more detailed description of the model.*

- Figure 1. Plotting $\lambda$ versus $\Delta GCox$ is not very intuitive to a broader audience. Though I recognize that these plots are useful, someone who is not experienced with these parameters will be confused. I recommend improving the clarity in visualizing the model output. I recommend complementing these plots (or substituting them) with something more recognized such as van Krevelen diagrams. I suggest plotting van Krevelens for river DOM, sediment DOM, and pyDOM and color coding the markers based on lambda and/or $\Delta GCox$. This way different compound classes can be easily identified in the H/C vs O/C space and we can see how energy content and metabolic efficiency vary per compound class (condensed molecules, phenols, lipids, etc.).

*We agree with the suggestion to add more common representations of data. We added Van Krevelen diagrams to Figure 1 and Figure S2.*

- It will be good to compare the model output \*computational\* results with previously published bio-degradation \*experimental\* studies. Bostick et al. 2021, Chen et al. 2022 and Norwood et al. 2013 provide degradation rate constants for pyDOM. Can the authors compute a similar bulk degradation rate constant using the output of their model and see how computational and experimental results compare? This will strengthen the conclusions of this manuscript.

*Thank you for the suggestion. The suggested papers report change or percent change in concentration through time, not a true rate constant (i.e., accounting for non-linearity through curve fitting). This discrepancy in units makes empirical comparisons difficult, and we have a strict 2,500 word limit. We dedicate a full subsection to comparisons with extant literature (2.3 Correspondence to Empirical Investigations., lines 205-232)*

Methods: Excellent text, but a lot of it reads like an introduction. Please move a lot of this text to other sections.

*Thank you for this suggestion. As per our response above, we have moved a portion of the methods into the introduction. Because the modelling approach we employed is relatively new and the*

*methods section does not have a word limit, we also retained much of the information in the methods to provide as much description as possible.*

Graphical Abstract: Absolutely gorgeous design. I would only recommend enhancing font sizes (difficult to read) and not using gray color – the labels (pyOM pool, Biomass, etc.) and other subfigures (e.g., biomass particles with tails) are difficult to read/see. I suggest just converting everything to bold black and increasing the sizes (just like CO2). Also why is "Biomass" looking like bacteria? I recommend replacing Biomass with "Microbes".

*Thank you! We increased the font size and adjusted the colors. The "biomass" is reflective of microbial biomass. We changed this label to "microbial biomass".*

Title: It is a representative title, but at present it reads a bit awkwardly. I read it as "bioavailability resembles DOM pools", which is odd. Consider rephrasing into something like "Computational modeling reveals that molecules in pyrogenic and natural dissolved organic matter pools have similar bio-lability" or something like that.

*We revised the title to provide more nuance: "Potential bioavailability of representative pyrogenic organic matter compounds in comparison to natural dissolved organic matter pools"*

References

Bostick, K.W., Zimmerman, A.R., Goranov, A.I., Mitra, S., Hatcher, P.G. and Wozniak, A.S. (2021) Biolability of fresh and photodegraded pyrogenic dissolved organic matter from laboratory-prepared chars. Journal of Geophysical Research: Biogeosciences 126, 1-17.

Chen, Y., Sun, K., Sun, H., Yang, Y., Li, Y., Gao, B. and Xing, B. (2022) Photodegradation of pyrogenic dissolved organic matter increases bioavailability: Novel insight into bioalteration, microbial community succession, and C and N dynamics. Chemical Geology, 120964.

Garayburu-Caruso, V.A., Danczak, R.E., Stegen, J.C., Renteria, L., Mccall, M., Goldman, A.E., Chu, R.K., Toyoda, J., Resch, C.T., Torgeson, J.M., Wells, J., Fansler, S., Kumar, S. and Graham, E.B. (2020) Using Community Science to Reveal the Global Chemogeography of River Metabolomes. Metabolites 10, 518.

Goranov, A.I., Wozniak, A.S., Bostick, K.W., Zimmerman, A.R., Mitra, S. and Hatcher, P.G. (2020) Photochemistry after fire: Structural transformations of pyrogenic dissolved organic matter elucidated by advanced analytical techniques. Geochimica et Cosmochimica Acta 290, 271-292.

Goranov, A.I., Wozniak, A.S., Bostick, K.W., Zimmerman, A.R., Mitra, S. and Hatcher, P.G. (2022) Microbial labilization and diversification of pyrogenic dissolved organic matter. Biogeosciences 19, 1491-1514.

Leyva, D., Tose, L.V., Porter, J., Wolff, J., Jaffé, R. and Fernandez-Lima, F. (2019) Understanding the structural complexity of dissolved organic matter: isomeric diversity. Faraday Discussions 218, 431-440.

McKenna, A.M., Chacón-Patiño, M.L., Chen, H., Blakney, G.T., Mentink-Vigier, F., Young, R.B., Ippolito, J.A. and Borch, T. (2021) Expanding the Analytical Window for Biochar Speciation: Molecular Comparison of Solvent Extraction and Water-Soluble Fractions of Biochar by FT-ICR Mass Spectrometry. Analytical Chemistry.

Smith, C.R., Hatcher, P.G., Kumar, S. and Lee, J.W. (2016) Investigation into the sources of biochar water-soluble organic compounds and their potential toxicity on aquatic microorganisms. ACS Sustainable Chemistry & Engineering 4, 2550-2558.

Stubbins, A., Spencer, R.G.M., Chen, H., Hatcher, P.G., Mopper, K., Hernes, P.J., Mwamba, V.L., Mangangu, A.M., Wabakanghanzi, J.N. and Six, J. (2010) Illuminated darkness: Molecular signatures of Congo River dissolved organic matter and its photochemical alteration as revealed by ultrahigh precision mass spectrometry. Limnology and Oceanography 55, 1467-1477.

Wagner, S., Ding, Y. and Jaffé, R. (2017) A new perspective on the apparent solubility of dissolved black carbon. Frontiers in Earth Science 5, 1-16.

Wagner, S., Harvey, E., Baetge, N., McNair, H., Arrington, E. and Stubbins, A. (2021) Investigating atmospheric inputs of dissolved black carbon to the Santa Barbara Channel during the Thomas Fire (California, USA). Journal of Geophysical Research: Biogeosciences n/a, e2021JG006442.

Ward, C.P., Sleighter, R.L., Hatcher, P.G. and Cory, R.M. (2014) Insights into the complete and partial photooxidation of black carbon in surface waters. Environmental Science: Processes & Impacts 16, 721-731.

Wozniak, A.S., Goranov, A.I., Mitra, S., Bostick, K.W., Zimmerman, A.R., Schlesinger, D.R., Myneni, S. and Hatcher, P.G. (2020) Molecular heterogeneity in pyrogenic dissolved organic matter from a thermal series of oak and grass chars. Organic Geochemistry 148, 1-18.

Yan, W., Chen, Y., Han, L., Sun, K., Song, F., Yang, Y. and Sun, H. (2022) Pyrogenic dissolved organic matter produced at higher temperature is more photoactive: Insight into molecular changes and reactive oxygen species generation. Journal of Hazardous Materials 425, 127817.

**Response to Reviewer 2**

General comments

In this manuscript Graham et al. investigate the bioavailability of pyrogenic organic matter (PyOM) using a substrate-explicit model, which is then compared to that of natural dissolved organic matter and water-extracted particulate organic matter. The current understanding of the impact of PyOM in freshwaters remains mainly speculative. On this note, the manuscript addresses an important topic in riverine biogeochemistry that would be of interest to the scientific community. The manuscript is also very well-written and easy to follow. I would recommend its publication after major revisions.

*Thank you for the constructive and positive comments.*

Specific comments

1. Based on the compounds selected as representation of PyOM, I wonder if there is any information in the literature regarding their experimental bioavailability. The same applies to DOM and POM. The authors could expand a bit more in the introduction to further clarify the contribution of the study they are presenting.

*Thank you for this comment. While empirical studies are relatively rare on PyOM bioavailability, there are several that exist (Norwood et al., 2013; Bostick et al., 2021; Chen et al., 2022). We discuss these studies at a high level on lines 59-68. We have a strict word limit of 2,500 words and we are currently at maximum length. While we are unable to add extensive details, we significantly revised the abstract and the last paragraph of the introduction to provide more clarity on the contribution of this paper.*

2. The rationale behind the experimental design is not completely clear and could not be adequate to test the proposed hypothesis. PyOM derived compounds mainly exhibit high Kow values that indicate their low solubility in water. In fact, some of the compounds included in Table S1 were determined after solvent extraction or CuO oxidation according to the references cited therein. However, these PyOM representative compounds were then compared to natural water-soluble organic matter (dissolved and particulate). Regardless, the authors report similar bioavailability parameters across phases, raising concerns about the model selection. This is because of the range of compounds with totally different chemical and physical properties that are being compared. I wonder why the list of PyOM derived compounds was not filtered to include just water-soluble compounds or the list of natural organic matter (dissolved and particulate) expanded to incorporate non-water-soluble compounds. These could represent an important overlooked fraction of natural organic matter, especially in the case of sediments. Also important is to include compounds that are detected beyond the 200-900 m/z analytical window or that escape the SPE procedure. I would recommend expanding the databases based on previously published literature and re-running the models. It would be interesting to see if similar results are obtained after expanding the composition of natural organic matter.

*Per this comment and the reviewer 1 comments above, we increased the database of PyOM molecules to over 16,000 molecules and found similar results. We also note the source of each molecule in Table S1.*

It is interesting that the authors included sediment water-extracted organic matter. This is usually not the rule in organic matter related studies in rivers, but definitely something that should be acknowledged more often.

*Thank you!*

4.      In the supplemental material, the authors mentioned that samples were normalised based on the concentration of dissolved organic carbon before SPE extraction. Given that the extraction efficiency of SPE cartridges is not constant, please add more information about how the organic matter extracts were normalised before FT-ICR-MS analysis or during data processing.

*Thank you for this comment. We note that no new FTICR-MS data was collected for this publication. While we recognize extraction efficiencies can vary by sample, the normalization procedure used by Garayburu-Caruso et al. was intended to standardize the amount of DOM passed through each filter, which can also lead to biases. This approach has been successfully employed across a variety of published literature and is the standard operating procedure at the Environmental Molecular Sciences Laboratory. Garayburu-Caruso et al. did not report the extraction efficiencies of individual SPE cartridges.*

*Textor, S.R., Wickland, K.P., Podgorski, D.C., Johnston, S.E. and Spencer, R.G., 2019. Dissolved organic carbon turnover in permafrost-influenced watersheds of interior Alaska: molecular insights and the priming effect. Frontiers in Earth Science, 7, p.275.*

*Stegen, J.C., Fansler, S.J., Tfaily, M.M., Garayburu-Caruso, V.A., Goldman, A.E., Danczak, R.E., Chu, R.K., Renteria, L., Tagestad, J. and Toyoda, J., 2022. Organic matter transformations are disconnected between surface water and the hyporheic zone. Biogeosciences, 19(12), pp.3099-3110.*

*Danczak, R.E., Goldman, A.E., Chu, R.K., Toyoda, J.G., Garayburu-Caruso, V.A., Tolić, N., Graham, E.B., Morad, J.W., Renteria, L., Wells, J.R. and Herzog, S.P., 2021. Ecological theory applied to environmental metabolomes reveals compositional divergence despite conserved molecular properties. Science of The Total Environment, 788, p.147409.*

5.      Please include information regarding quality controls used during FT-ICR-MS analysis.

*We note that no new FTICR-MS data was collected for this publication. Suwannee River Fulvic Acid (SRFA) was used as a quality control check in each run in the work reported by Garayburu-Caruso et al. This information is included in the supplemental methods.*

6.      I would strongly suggest using the ranges proposed by Laszakovits & MacKay (2022) to assign compound classes via van Krevelen diagrams (DOI: 10.1021/jasms.1c00230). Please update.

*There have been several iterations of van Krevelen classes proposed in the literature since the original citation, and there is debate surrounding the optimal classification system. Since we did*

*not generate any new FTICR-MS in this publication, we chose to use the classes assigned by Garayburu-Caruso et al. In response to this comment and reviewer 1, we added van Krevelen plots to Figure 1 and provide an addition figure showing unique molecules in each dataset (Figure S2). One of our intents with these additions is to allow readers to examine alternate classification thresholds. We also added two additional van Krevelen classifications for each molecule in the supplemental tables.*

7.      Please include the F-value of the results of the statistical analysis, when appropriate, in the main body or as supplemental material.

*Thank you. We added F-values as appropriate.*

8.      I would recommend that the authors include a statement in the *Conclusions* addressing their previously proposed hypothesis.

*We added a statement regarding our hypothesis at the beginning of the conclusions section. Lines 235-237: "Our work supports the hypothesis that PyOM may have similar overall bioavailability as compared to natural sources of DOM and provides a foundation for targeted experiments investigating specific components of the PyOM continuum."*

9.      The authors stated the limitations of this approach well enough (e.g., lines 226-229). This is important considering the implications and future work.

*Thank you!*

Technical corrections

line 53: please convert to Tg or Gg or an appropriate standard unit.

*Done.*

line 103: please use an appropriate notation (instead of p (PyOM-sediment)).

*We are unsure what change is being suggested. We are happy to alter the notation if the reviewer can provide an example.*

line 300: Is the dataset in Garayburu-Caruso et al. (2020a) the most comprehensive assessment of DOM in rivers to date?

*We agree this was unnecessary. We removed this wording from the manuscript.*

line 331: please include the references for the R software as well as for each package.

*Done.*